# Comparative Genomic Analysis of *Fusobacterium necrophorum* Provides Insights into Conserved Virulence Genes

Prabha K. Bista,[a] Deepti Pillai,[a,b] Chayan Roy,[c,d] Joy Scaria,[c] Sanjeev K. Narayanan[a]

aDepartment of Comparative Pathobiology, Purdue University, West Lafayette, Indiana, USA
bIndiana Animal Disease and Diagnostic Laboratory, Purdue University, West Lafayette, Indiana, USA
cDepartment of Veterinary and Biomedical Sciences, South Dakota State University, Brookings, South Dakota, USA
dEnvironment Microbial Genomics, Plant and Environmental Microbiology, Copenhagen University, Copenhagen, Denmark

**ABSTRACT** *Fusobacterium necrophorum* is a Gram-negative, filamentous anaerobe prevalent in the mucosal flora of animals and humans. It causes necrotic infections in cattle, resulting in a substantial economic impact on the cattle industry. Although infection severity and management differ within *F. necrophorum* species, little is known about *F. necrophorum* speciation and the genetic virulence determinants between strains. To characterize the clinical isolates, we performed whole-genome sequencing of four bovine isolates (8L1, 212, B17, and SM1216) and one human isolate (MK12). To determine the phylogenetic relationship and evolution pattern and investigate the presence of antimicrobial resistance genes (ARGs) and potential virulence genes of *F. necrophorum*, we also performed comparative genomics with publicly available *Fusobacterium* genomes. Using up-to-date bacterial core gene (UBCG) set analysis, we uncovered distinct *Fusobacterium* species and *F. necrophorum* subspecies clades. Pangenome analyses revealed a high level of diversity among *Fusobacterium* strains down to species levels. The output also identified 14 and 26 genes specific to *F. necrophorum* subsp. *necrophorum* and *F. necrophorum* subsp. *funduliforme*, respectively, which could be essential for bacterial survival under different environmental conditions. ClonalFrameML-based recombination analysis suggested that extensive recombination among accessory genes led to species divergence. Furthermore, the only strain of *F. necrophorum* with ARGs was *F. necrophorum* subsp. *funduliforme* B35, with acquired macrolide and tetracycline resistance genes. Our custom search revealed common virulence genes, including toxins, adhesion proteins, outer membrane proteins, cell envelope, type IV secretion system, ABC (ATP-binding cassette) transporters, and transporter proteins. A focused study on these genes could help identify major virulence genes and inform effective vaccination strategies against fusobacterial infections.

**IMPORTANCE** *Fusobacterium necrophorum* is an anaerobic bacterium that causes liver abscesses in cattle with an annual incidence rate of 10% to 20%, resulting in a substantial economic impact on the cattle industry. The lack of definite biochemical tests makes it difficult to distinguish *F. necrophorum* subspecies phenotypically, where genomic characterization plays a significant role. However, due to the lack of a good reference genome for comparison, *F. necrophorum* subspecies-level identification represents a significant challenge. To overcome this challenge, we used comparative genomics to validate clinical test strains for subspecies-level identification. The findings of our study help predict specific clades of previously uncharacterized strains of *F. necrophorum*. Our study identifies both general and subspecies-specific virulence genes through a custom search-based analysis. The virulence genes identified in this study can be the focus of future studies aimed at evaluating their potential as vaccine targets to prevent fusobacterial infections in cattle.

**KEYWORDS** *Fusobacterium necrophorum*, virulence genes, phylogeny, pangenome, recombination, virulence factors, subspecies

Address correspondence to Sanjeev K. Narayanan, sanjeev@purdue.edu.

The authors declare no conflict of interest.

The genus *Fusobacterium* is a Gram-negative, anaerobic, non-spore-forming bacillus commonly found in the oral, intestinal, respiratory, and genital tracts of animals and humans. *Fusobacterium* produces volatile fatty acids, including butyrate, acetate, and propionate, as major metabolic end products (1). Of the many species in this genus, *Fusobacterium necrophorum* is an important opportunistic pathogen in both human and veterinary medicine. In humans, *F. necrophorum* is a causative agent in a variety of localized necrotic throat infections, such as tonsillitis, and systemic infections, such as Lemierre's syndrome (2, 3). In animals, it causes severe cases of calf diphtheria, liver abscesses, and foot rot (4–7). As a consequence, *F. necrophorum* is responsible for substantial economic losses in the feedlot industry (8). *F. necrophorum* is divided into two main subspecies, *F. necrophorum* subsp. *necrophorum* (biotype A) and *F. necrophorum* subsp. *funduliforme* (biotype B), based on differences in cellular morphology, biochemical characterization, and differences in DNA level in 16S rRNA and DNA repeats (9–12). *F. necrophorum* subsp. *necrophorum* is more virulent and has thus been isolated more frequently from infections than *F. necrophorum* subsp. *funduliforme* (13).

Complete *F. necrophorum* subsp. *funduliforme* genome sequences have been published in recent years. However, the genome sequences of *F. necrophorum* subsp. *necrophorum* are not well characterized. Although there are a few additional genomes of *F. necrophorum* publicly available, the subspecies have not been properly defined. The lack of reference genome sequences makes the classification and phylogeny of *F. necrophorum* particularly challenging. Because there is no established biochemical methodology for subspecies differentiation, the phenotypic characterization of *F. necrophorum* is exclusively dependent on the output of the Rapid Ana II system (14). Consequently, genomic characterization of these clinical isolates could help validate the phenotypic characterizations. Also, to determine the phylogenetic relationship and evolution pattern and explore potential virulence genes of *F. necrophorum*, we genetically characterized five clinical isolates of *F. necrophorum* and confirmed the phenotypic representation of those isolates using genetic approaches. This study adds more reference genomes for additional *F. necrophorum* subspecies.

With rapid advances in sequencing methods, whole-genome sequencing technology has been widely used to predict pathogens' phenotypic traits, such as virulence and antimicrobial resistance. Sequence analysis offers better insight into strain-specific genetic variation, disease propensity, phylogenetic relationships, virulence patterns, potential virulence genes, and diversity among subspecies and strains (15, 16). This critical information is lacking for *F. necrophorum*, an opportunistic pathogen that is one of the main causes of liver abscesses (17). Therefore, in this study, we used whole-genome sequence analysis to understand *F. necrophorum* virulence by determining the genetic content of virulence and antimicrobial resistance genes (ARGs). We concentrated on understanding the phylogenetic relationship, evolution pattern, and potential virulence genes of *F. necrophorum* at the subspecies level.

## RESULTS

**Phylogenetic analyses of *Fusobacterium* and *F. necrophorum*. (i) Phylogenetic distribution and relationship of *Fusobacterium* genus.** We studied genomic evolution by assembling dissimilarity matrices within the core genomes of 167 *Fusobacterium* strains (162 latest [July 2019] RefSeq sequences plus five test strains) and illustrated the results in Fig. 1A based on phylogenetic analyses of the 92 core genes (up-to-date bacterial core genes [UBCGs]). Two distinct strains, *Leptotrichia buccalis* and *Cetobacterium somerae*, were used as an outgroup (Fig. 1A, red font).

The data revealed eight distinct clades of *Fusobacterium* species. In most clades, the *Fusobacterium* species within a clade shown in Fig. 1A with a given color belonged to the same species. In particular, *F. necrophorum* (Fig. 1A, red) formed a distinct clade, suggesting a close relationship with the species *Fusobacterium gonidiaformans*. The genome with the most sequences available, *Fusobacterium nucleatum* (Fig. 1A, green), formed a distinct clade. Other representative clades of *Fusobacterium* species included *Fusobacterium hwasooki* (Fig. 1A, pink), *F. periodonticum* (yellow), *F. mortiferum* (dark

# A

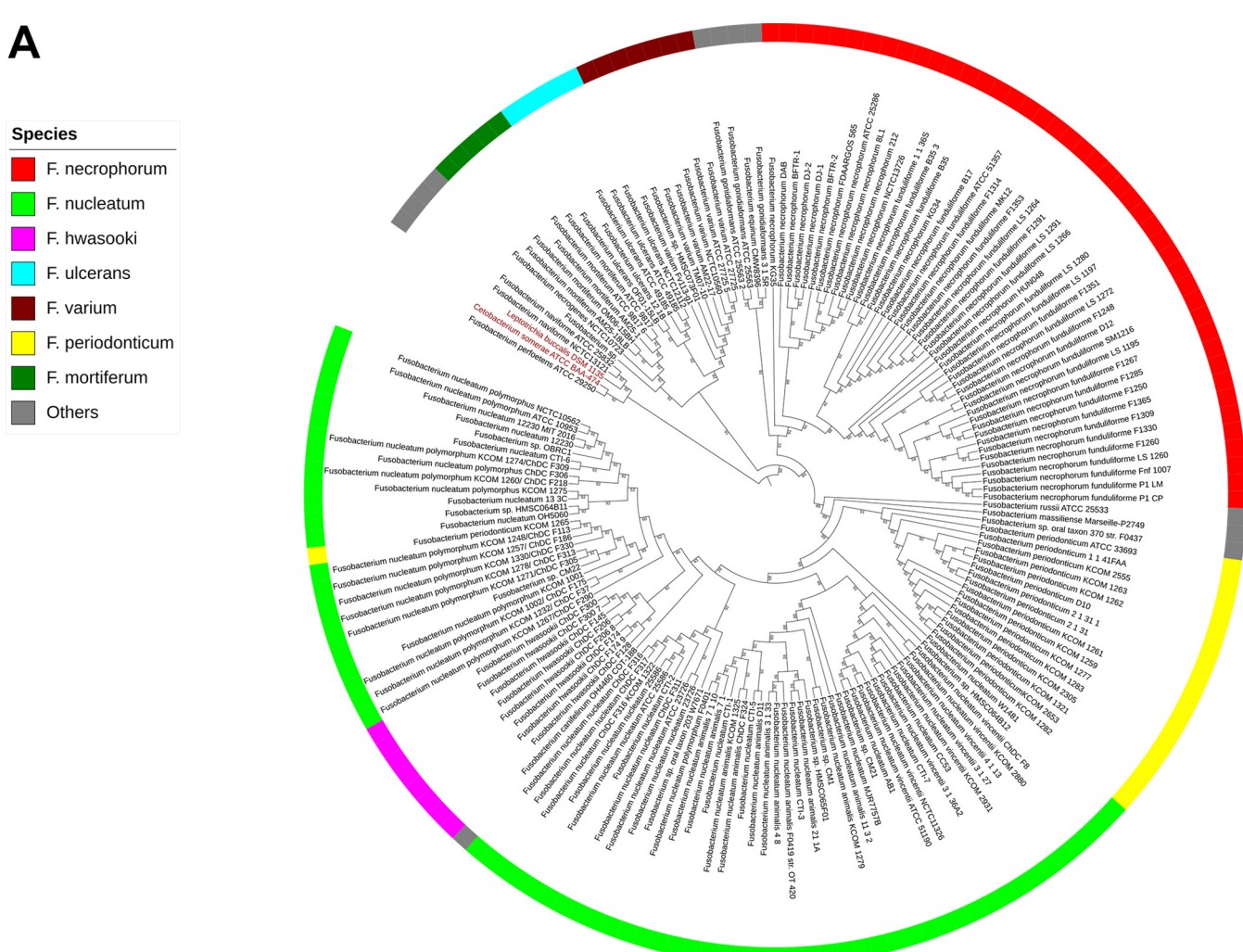

**FIG 1** (A) Gene content cladogram of genus *Fusobacterium*. The unweighted pair group method with arithmetic means (UPGMA) cladogram was constructed based on the dissimilarities among core gene sequence analyses using a UBCG set in 167 *Fusobacterium* strains. *Leptotrichia buccalis* DSM 1135 and *Cetobacterium somerae* ATCC BAA-474 (red font) were used as outgroups. Each colored strip denotes a different *Fusobacterium* species, as defined in the legend. Gene support index values are given at branching points. (B) Phylogenetic tree of *Fusobacterium necrophorum* species. Maximum-likelihood phylogenetic tree based on UBCG sequences of 41 RefSeq and five *F. necrophorum* test strains (bold). The clades are distinctly defined for each subspecies. The blue branch represents the clade for *F. necrophorum* subsp. *necrophorum*, and the red branch represents the clade for *F. necrophorum* subsp. *funduliforme*. strains BL and FDAARGOS 565 in the red boxes fall into the clade of *F. necrophorum* subsp. *necrophorum*, indicating that they are most likely *F. necrophorum* subsp. *necrophorum*. Bar, 0.01substitution per nucleotide position. (C) Heat map of ANI of whole-genome comparison of 46 (41 RefSeq sequences and five test strains) *F. necrophorum* strains. The row and column labels in the heat map correspond to the pairwise alignment of 46 *F. necrophorum* species. Cells in the heat map correspond to 95% identity and 70% coverage or greater. The identity match ranges from green to yellow to red as identity matches drop to 94.5%. *F. necrophorum* subsp. *necrophorum* and *F. necrophorum* subsp. *funduliforme* form each group represented in two distinct green blocks in the heat map and clustered together into separate clades in the phylogenetic tree. KG35 had a comparatively small ANI value, with the lowest-identity matches (some values were <95%), as indicated in the red grid. Strains are represented by their acronym and are provided in a supplementary material S1, see Data set S1.

green), *F. ulcerans* (blue), and *F. varium* (dark red). Furthermore, the data revealed that several uncategorized *Fusobacterium* species (*F. naviforme*, *F. perfoetens*, *F. gonidiaformans*, *F. russii*, *F. massiliense*, and *F. canifelinum*) formed a separate clade (Fig. 1A, gray). In the resulting cladogram, *F. naviforme* and *F. perfoetens* ATCC 29250 were closest to the outgroup strains and belonged to the same clade.

**(ii) Phylogenic relationships among *Fusobacterium necrophorum* subspecies.** We used the whole-genome sequence to perform phylogenetic analysis based on a concatenated bacterial core gene set in a UBCG pipeline. The phylogeny implemented using RAxML revealed that all test strains (B17, SM1216, MK12, 8L1, and 212) clustered according to their respective subspecies (i.e., the *F. necrophorum* subsp. *funduliforme* and *F. necrophorum* subsp. *necrophorum* clades) (Fig. 1B). These findings cross-confirmed the phenotypic classification of the test strains. However, some of the RefSeq species in

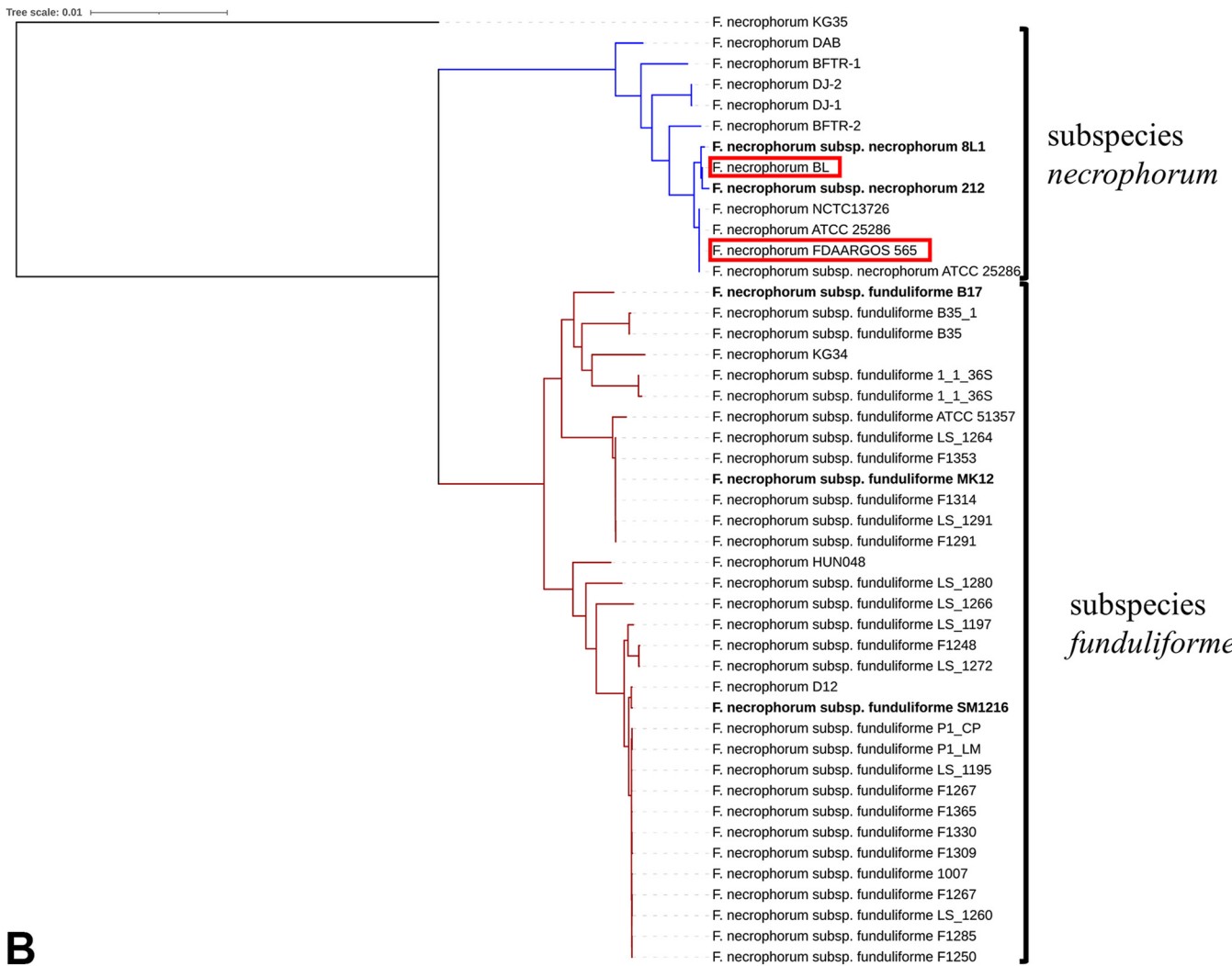

**B**

**FIG 1** (Continued)

this study have not been identified to the subspecies level. These RefSeq species were placed in *F. necrophorum* subsp. *necrophorum* or *F. necrophorum* subsp. *funduliforme* on the phylogenetic tree based on the core genome (Fig. 1B), except for strain *F. necrophorum* KG35. Table 1 shows the predicted subspecies of these RefSeq strains based on the phylogenetic tree (Fig. 1B). For example, the RefSeq strains *F. necrophorum* BL and *F. necrophorum* FDAARGOS_565 clustered with the *F. necrophorum* subsp. *necrophorum* ATCC 25286 (the only complete published *F. necrophorum* subsp. *necrophorum* genome) and test strains 8L1 and 212 (phenotypically identified as *F. necrophorum* subsp. *necrophorum*). Therefore, UBCG analysis showed that out of 46 strains, 12 strains belonged to *F. necrophorum* subsp. *necrophorum*, while 34 belonged to *F. necrophorum* subsp. *funduliforme*.

(iii) **Average nucleotide identity analysis.** To confirm our phylogenetic analysis, we also used average nucleotide identity (ANI) analysis, which gives the mean of nucleotide identity values for *F. necrophorum* strains. Using the FastANI results, we calculated pairwise ANI values among the *F. necrophorum* strains, which ranged from 94.64% to 99.99%. Next, we generated a heat map to visualize RefSeq and test strain clustering based on the distance matrix of ANI values (Fig. 1C). Each cell in the heat map corresponds to 95% identity and 70% coverage (indicating the same subspecies). As the comparison approaches 94.5%, the intensity of the color changes from green (>97% match) to yellow (95% to 97% match) to red (94% to 95% match). Within the heat map, there are defined clusters that separate

# C

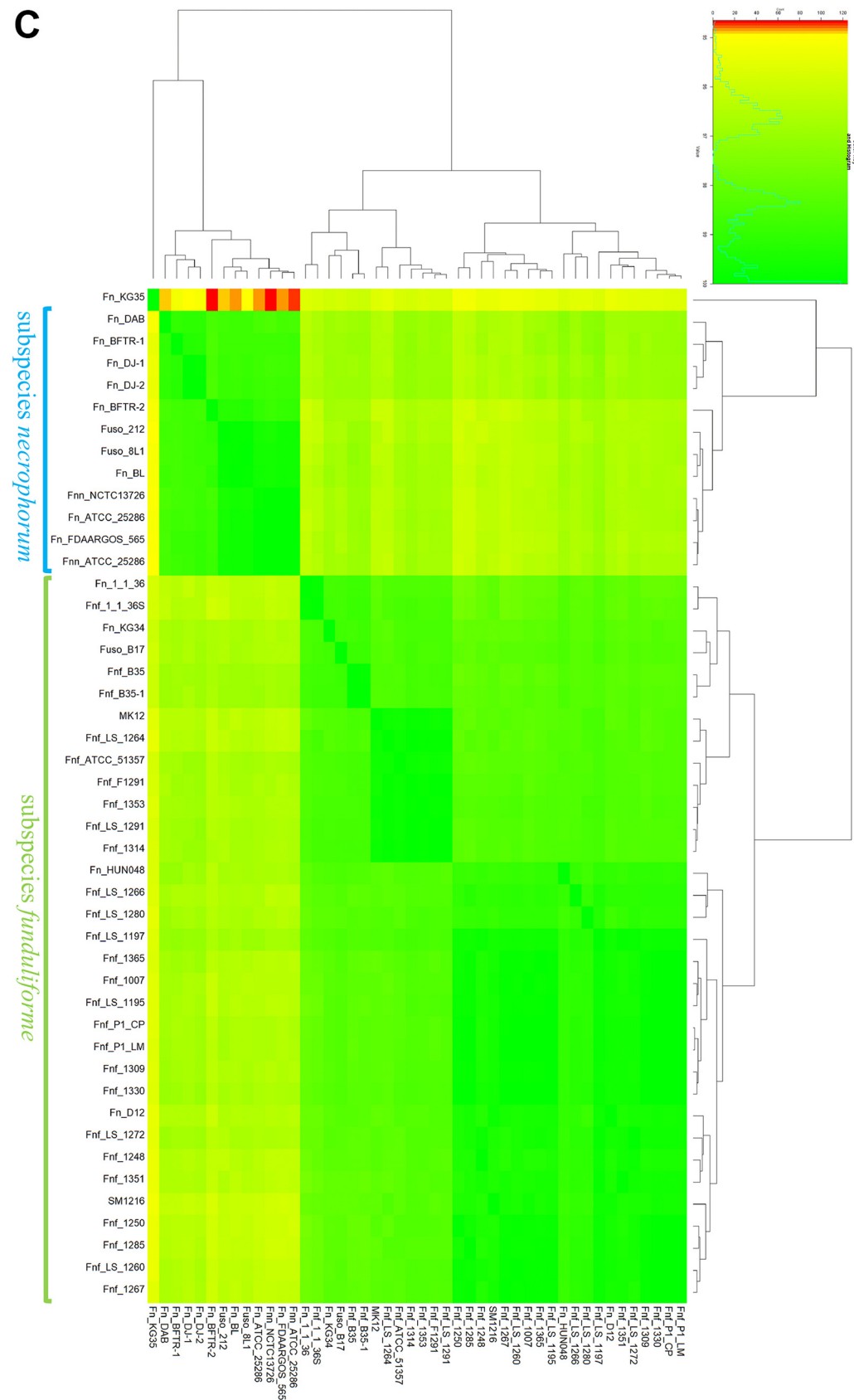

**FIG 1** (Continued)

**TABLE 1** Predicted subspecies of publicly available reference sequences of *Fusobacterium necrophorum* based on phylogenetic analysis

| Species, subspecies, and strain | Predicted subspecies | Type | Source of isolation[a] |
|---|---|---|---|
| *F. necrophorum* subsp. *funduliforme* ATCC 51357 | *funduliforme* | RefSeq | NA |
| *F. necrophorum* subsp. *funduliforme* 1_1_36S | *funduliforme* | RefSeq | NA |
| *F. necrophorum* subsp. *funduliforme* F1291 | *funduliforme* | RefSeq | Blood |
| *F. necrophorum* subsp. *funduliforme* F1260 | *funduliforme* | RefSeq | Blood |
| *F. necrophorum* FDAARGOS_565 | *necrophorum* | RefSeq | Environmental |
| *F. necrophorum* subsp. *necrophorum* ATCC 25286 | *necrophorum* | RefSeq | NA |
| *F. necrophorum* D12 | *funduliforme* | RefSeq | NA |
| *F. necrophorum* HUN048 | *funduliforme* | RefSeq | NA |
| *F. necrophorum* subsp. *funduliforme* 1007 | *funduliforme* | RefSeq | NA |
| *F. necrophorum* subsp. *funduliforme* B35 | *funduliforme* | RefSeq | Bovine liver abscess |
| *F. necrophorum* BL | *necrophorum* | RefSeq | Bovine liver abscess |
| *F. necrophorum* DJ-1 | *necrophorum* | RefSeq | Deer tongue/jaw |
| *F. necrophorum* BFTR-1 | *necrophorum* | RefSeq | Bovine foot rot |
| *F. necrophorum* DAB | *necrophorum* | RefSeq | Deer jaw abscess |
| *F. necrophorum* BFTR-2 | *necrophorum* | RefSeq | Bovine foot rot |
| *F. necrophorum* DJ-2 | *necrophorum* | RefSeq | Deer jaw |
| *F. necrophorum* subsp. *funduliforme* B35 | *funduliforme* | RefSeq | Bovine liver abscess |
| *F. necrophorum* subsp. *funduliforme* LS_1260 | *funduliforme* | RefSeq | Human blood |
| *F. necrophorum* subsp. *funduliforme* LS_1264 | *funduliforme* | RefSeq | Human blood |
| *F. necrophorum* subsp. *funduliforme* LS_1197 | *funduliforme* | RefSeq | Human blood |
| *F. necrophorum* subsp. *funduliforme* LS_1195 | *funduliforme* | RefSeq | Human blood |
| *F. necrophorum* subsp. *funduliforme* LS_1266 | *funduliforme* | RefSeq | Human blood |
| *F. necrophorum* subsp. *funduliforme* LS_1280 | *funduliforme* | RefSeq | Human blood |
| *F. necrophorum* subsp. *funduliforme* LS_1272 | *funduliforme* | RefSeq | Human blood |
| *F. necrophorum* subsp. *funduliforme* LS_1291 | *funduliforme* | RefSeq | Human blood |
| *F. necrophorum* subsp. *funduliforme* F1248 | *funduliforme* | RefSeq | Human throat swab |
| *F. necrophorum* subsp. *funduliforme* F1285 | *funduliforme* | RefSeq | Human throat swab |
| *F. necrophorum* subsp. *funduliforme* F1250 | *funduliforme* | RefSeq | Human throat swab |
| *F. necrophorum* subsp. *funduliforme* F1267 | *funduliforme* | RefSeq | Human throat swab |
| *F. necrophorum* subsp. *funduliforme* F1309 | *funduliforme* | RefSeq | Human throat swab |
| *F. necrophorum* subsp. *funduliforme* F1314 | *funduliforme* | RefSeq | Human throat swab |
| *F. necrophorum* subsp. *funduliforme* F1353 | *funduliforme* | RefSeq | Human throat swab |
| *F. necrophorum* subsp. *funduliforme* F1330 | *funduliforme* | RefSeq | Human throat swab |
| *F. necrophorum* subsp. *funduliforme* F1365 | *funduliforme* | RefSeq | Human throat swab |
| *F. necrophorum* subsp. *funduliforme* P1_CP | *funduliforme* | RefSeq | Colorectal tumor tissue |
| *F. necrophorum* subsp. *funduliforme* P1_LM | *funduliforme* | RefSeq | Liver metastasis tumor tissue |
| *F. necrophorum* KG34 | *funduliforme* | RefSeq | Cow uterine swab |
| *F. necrophorum* ATCC 25286 | *necrophorum* | RefSeq | Cow uterine swab |
| *F. necrophorum* subsp. *funduliforme* 1_1_36S | *funduliforme* | RefSeq | NA |
| *F. necrophorum* subsp. *necrophorum* NCTC13726 | *necrophorum* | RefSeq | Bovine liver abscess |
| *F. necrophorum* subsp. *necrophorum* 8L1 | *necrophorum* | WGS test | Bovine liver abscess |
| *F. necrophorum* subsp. *necrophorum* 212 | *necrophorum* | WGS test | Bovine foot rot |
| *F. necrophorum* subsp. *funduliforme* B17 | *funduliforme* | WGS test | Bovine liver abscess |
| *F. necrophorum* subsp. *funduliforme* MK12 | *funduliforme* | WGS test | Human tonsil |
| *F. necrophorum* subsp. *funduliforme* SM1216 | *funduliforme* | WGS test | Bovine foot rot |

[a]NA, not available.

clades for *F. necrophorum* subsp. *funduliforme* from those for *F. necrophorum* subsp. *necrophorum*.

All ANI values between *F. necrophorum* strains (including all test strains) were ≥95%, suggesting that the strains belong to *F. necrophorum*. However, the heat map showed similarities ranging from 94.645% to 96.028% for *F. necrophorum* KG35. The homology with ANI values <95% between the *F. necrophorum* strains KG35 and BFTR-2 (94.688%), KG35 and ATCC 25286 (94.739%), and KG35 and NCTC13726 (94.645%) is represented in the red grid in the heat map (Fig. 1C).

**Pangenome analyses.** To understand the genomic diversity among *F. necrophorum* strains, we performed pangenome analyses to compare core and accessory genes. We performed pangenome analysis on 167 *Fusobacterium* strains (RefSeq strains plus five test strains), and we included two outgroups to evaluate genus-level diversity. The *Fusobacterium* genus pangenome contained 52,441 gene clusters (Fig. 2A, part 1). Of

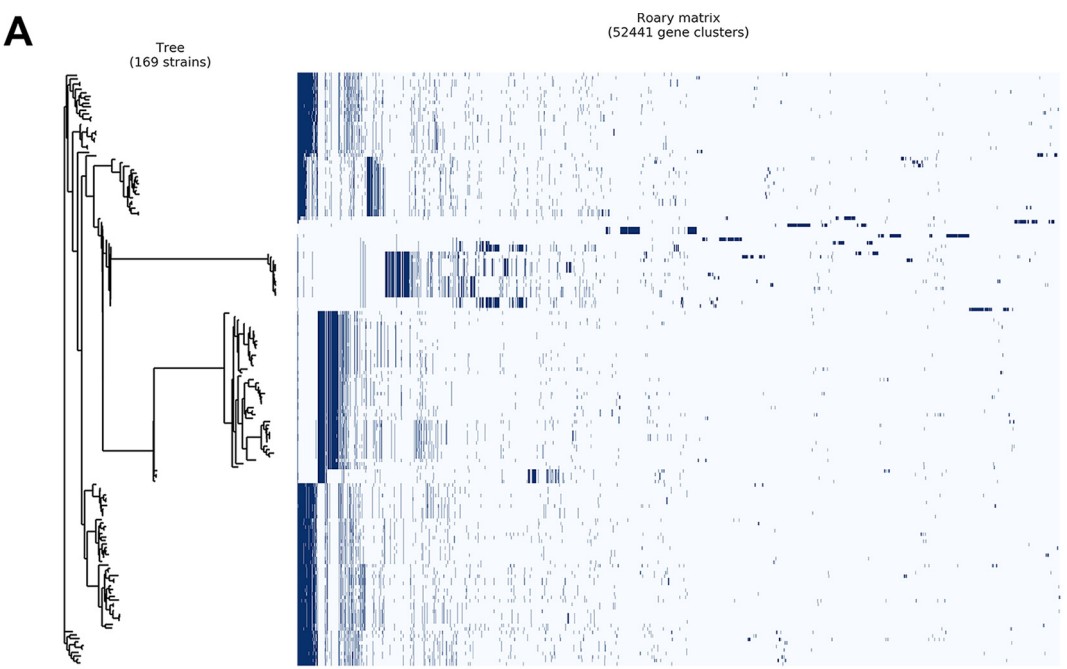

1. *Fusobacterium* strains

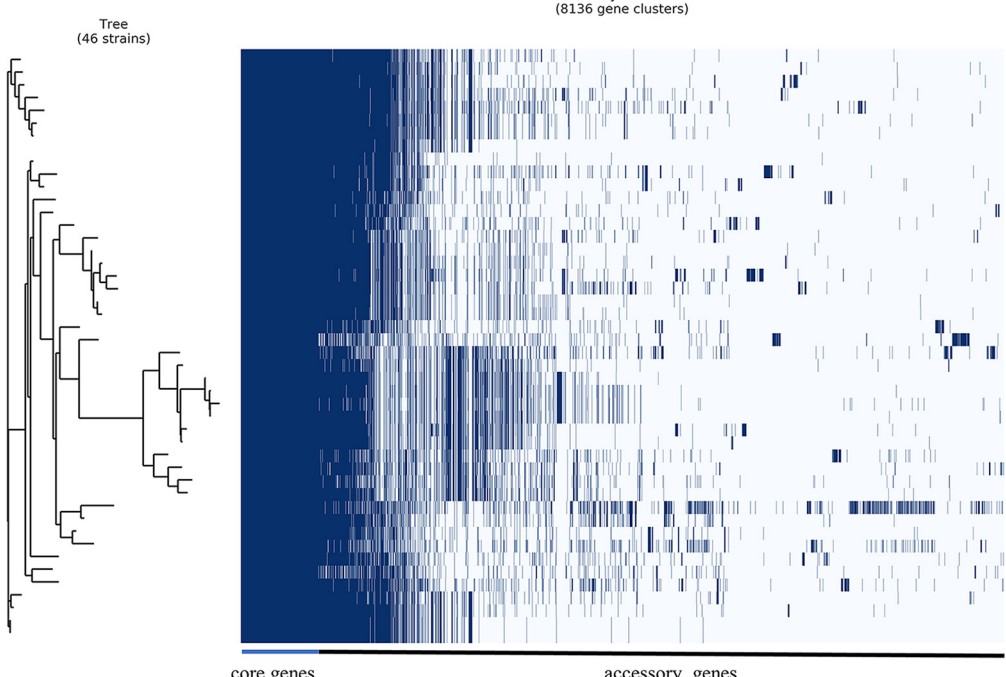

core genes              accessory genes

2. *Fusobacterium necrophorum* strains

FIG 2 (A) Pangenome alignment of *Fusobacterium* species. Pangenome analysis was performed using Roary. Pangenome analysis of *Fusobacterium* (genus level [part 1]) and *F. necrophorum* (species level [part 2]). The gene matrix shows the presence (blue blocks) and absence (gray areas) of core and accessory genes. In the matrix, genomes are shown as rows and gene clusters as columns. The dendrogram is based on the phylogenetic relatedness of the core genes and accessory gene clusters. (B) *F. necrophorum* population structure cladogram. The maximum-likelihood tree was constructed based on the pangenome analysis with 826 core genes and calculated based on the GTR+I+G4 model and 100 bootstrap replicates for 46 *F. necrophorum* strains. The right side of the cladogram includes three color-coded columns representing GC content, source, and genome size of the isolates. The cladogram forms distinct clades of *F. necrophorum* subsp. *funduliforme*, *F. necrophorum* subsp. *necrophorum*, and *F. necrophorum* not identified to the subspecies level. (C) *F. necrophorum* subsp. *necrophorum*- and *F. necrophorum* subsp. *funduliforme*-specific genes. Genes specific to *F. necrophorum* subsp. *necrophorum* and *F. necrophorum* subsp. *funduliforme* were separated by functional annotation. In the grid, color represents the presence and the absence of color shows the absence of genes. A total of 40 subspecies-specific genes with different biological functions were identified through the UniProt-based search. Nineteen hypothetical genes identified are shown in yellow.

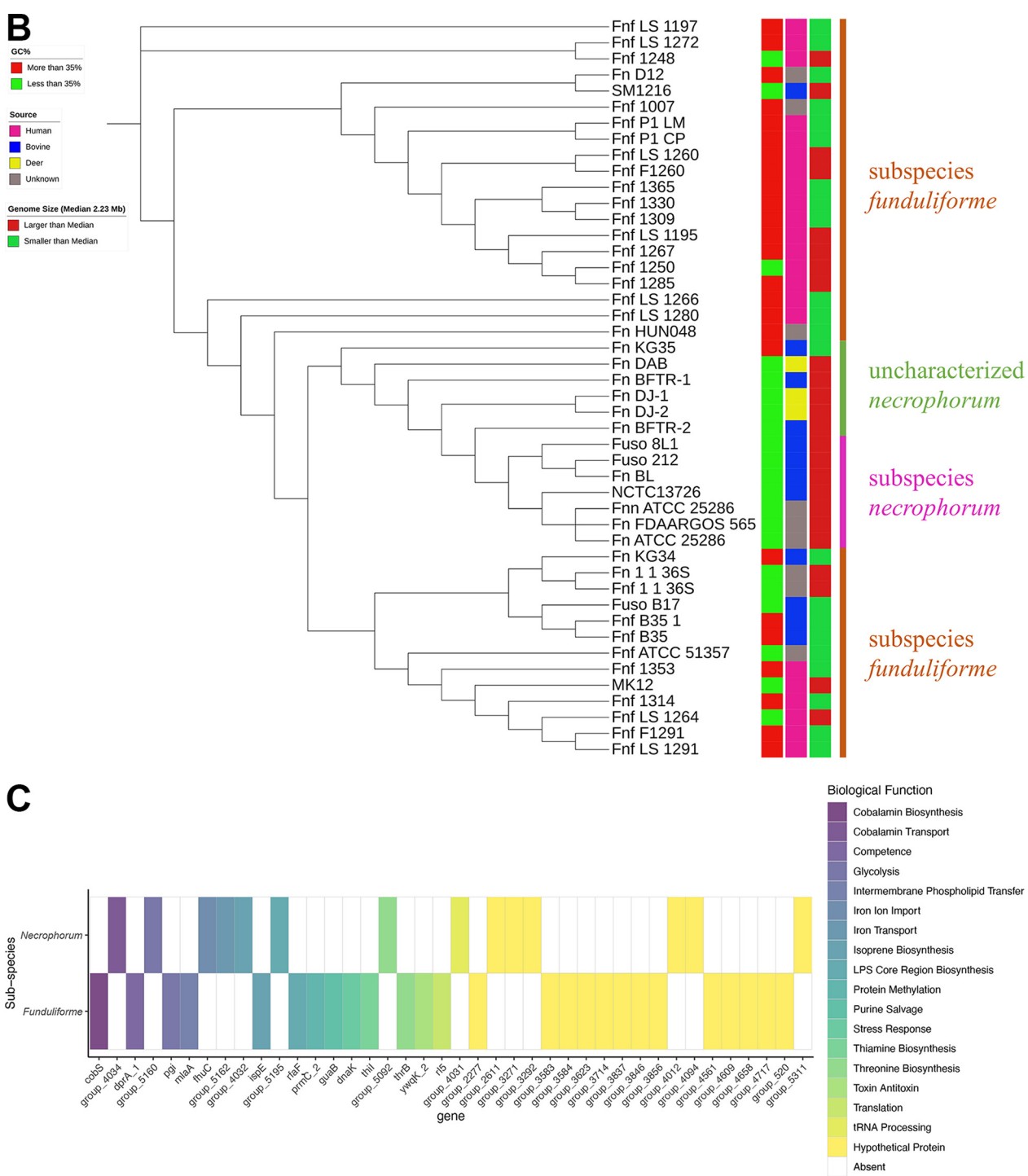

**FIG 2** (Continued)

the 52,441 gene clusters, 29 (0.05%) constituted the core genes and 2 (0.004%) the soft core genes. There were 3,855 and 42,332 shell and cloud genes, respectively, indicating that 88.07% of the genes in the genome are accessory genes, suggesting that every species diverged enormously and formed new phenotypic traits. The set of 29 core genes included genes encoding cellular translation proteins, such as ribosomal proteins, and enzyme-encoding genes (see Data Set S7 in the supplemental material). Most of these core genes were also included in UBCG sets.

Next, we performed pangenome analysis on 46 *F. necrophorum* strains (41 NCBI RefSeq strains plus five test strains) to compare unique genes present at the species level. This pangenome contained 8,136 gene clusters. Of these 8,136 gene clusters, a total of 826 (10.15%) constituted the core genes and 404 (0.0004%) the soft core genes. Notably, the proportions of shell and cloud genes were higher, 1,856 (22.81%) and 5,050 (62.06%), respectively. Using whole-genome alignment with comparisons of core and ancillary genes (generated using Roary), we examined the genetic relatedness of the 46 strains (Fig. 2A, part 2). We made further comparisons by clustering *F. necrophorum* species based on the alignment of 826 core genes obtained from the 46 *F. necrophorum* strains (Fig. 2B). In the resulting cladogram, test strains B17, MK12, and SM1216 fell into the *F. necrophorum* subsp. *funduliforme* clade. Likewise, test strains 8L1 and 212 formed a separate clade along with some of the unspecified *F. necrophorum* strains and the known *F. necrophorum* subsp. *necrophorum*. These findings demonstrate that *F. necrophorum* subsp. *necrophorum* and *F. necrophorum* subsp. *funduliforme* formed a distinct clade (Fig. 2B).

Based on the pangenome-wide association analysis, *F. necrophorum* subsp. *funduliforme* had 26 subspecies-specific genes and *F. necrophorum* subsp. *necrophorum* has 14 subspecies-specific genes. The list of genes with their biological functions is provided in Data Set S4. The subspecies-level unique genes are shown in a binary matrix group and defined by their biological function in Fig. 2C. We observed differences in the biological functional group between the subspecies. These included genes that encode transferases, transporters, an antitoxin YwqK in *F. necrophorum* subsp. *funduliforme* and enzymes for various pathways such as vitamin biosynthetic (thiamine and cobalamin) and metabolic pathways (transferases and dehydrogenases). There are some unique genes in *F. necrophorum* subsp. *necrophorum*. These genes are for iron transport and biosynthetic pathways. Overall, 19 of 40 genes have not been annotated and are labeled as hypothetical genes.

**Distribution of virulence genes and antimicrobial resistance genes. (i) Virulence genes.** Because the widely used virulence factor database did not have the *Fusobacterium* reference genome (18), we created a custom database with 46 potential virulence genes to analyze. This database was used to determine potential virulence genes in *F. necrophorum* RefSeq strains and the five test strains. Based on a customized search approach, we identified 13 different virulence genes (Fig. 3A). These virulence genes encoded the following traits: toxins (leukotoxin and hemolysin), adhesion proteins (FadA, YadA C-terminal domain-containing protein, and trimeric autotransporter adhesins), outer membrane proteins (OmpA and -H), invasive capability (VapD), cell envelope (CreD), type IV secretion system, ATP-binding cassette (ABC) transporters, hemolysin activation protein (HecB), and transporter proteins (ShlB and FhaC). The presence of these virulence genes in the test strains is shown in Fig. 3A, and percent sequence similarity is shown in Data Set S6.

In particular, several virulence genes, including those for the type IV secretion system, ABC transporter system, and FadA adhesion protein, were unique to *F. necrophorum* subsp. *necrophorum* and not present in *F. necrophorum* subsp. *funduliforme*. All phenotypically characterized *F. necrophorum* subsp. *necrophorum* strains, including *F. necrophorum* BL, *F. necrophorum* FDAARGOS_565, *F. necrophorum* ATCC 25286, and test strains 8L1 and 212, clustered in the same clade in the phylogenetic tree (Fig. 1B, UBCG tree). A notable exception was one of the RefSeq strains, NCTC13726, as the type IV secretion system was absent despite the strain's belonging to the same *F. necrophorum* subsp. *necrophorum* clade.

Similarly, some virulence genes, such as genes encoding OmpA and Fic toxin, were present in all except *F. necrophorum* subsp. *necrophorum*. Furthermore, our genomic analyses identified specific virulence genes, including leukotoxin, hemolysin, *ompH*, *vapD*, and *creD*, that are present in almost all species of *F. necrophorum*.

**(ii) Antimicrobial resistance genes.** To identify the presence of ARGs, we screened both the RefSeq sequences and the test strains using ABRicate. This prediction tool identified several acquired ARGs in 38 *Fusobacterium* genomes, and the list of *Fusobacterium* species with different ARGs harbored by them is shown in Fig. 3B. We detected 19 acquired ARGs in the tested *Fusobacterium* strains (Fig. S1). A total of 38 genomes carried at least one of the 19 acquired ARGs available in the ABRicate databases. Most of these

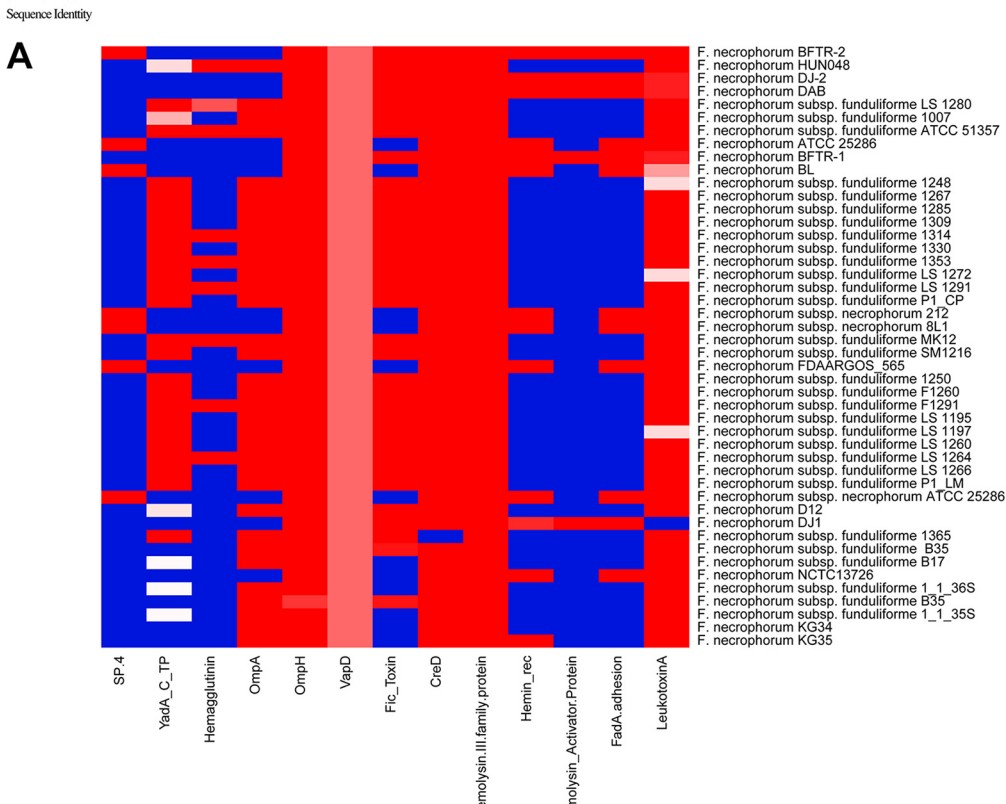

**A**

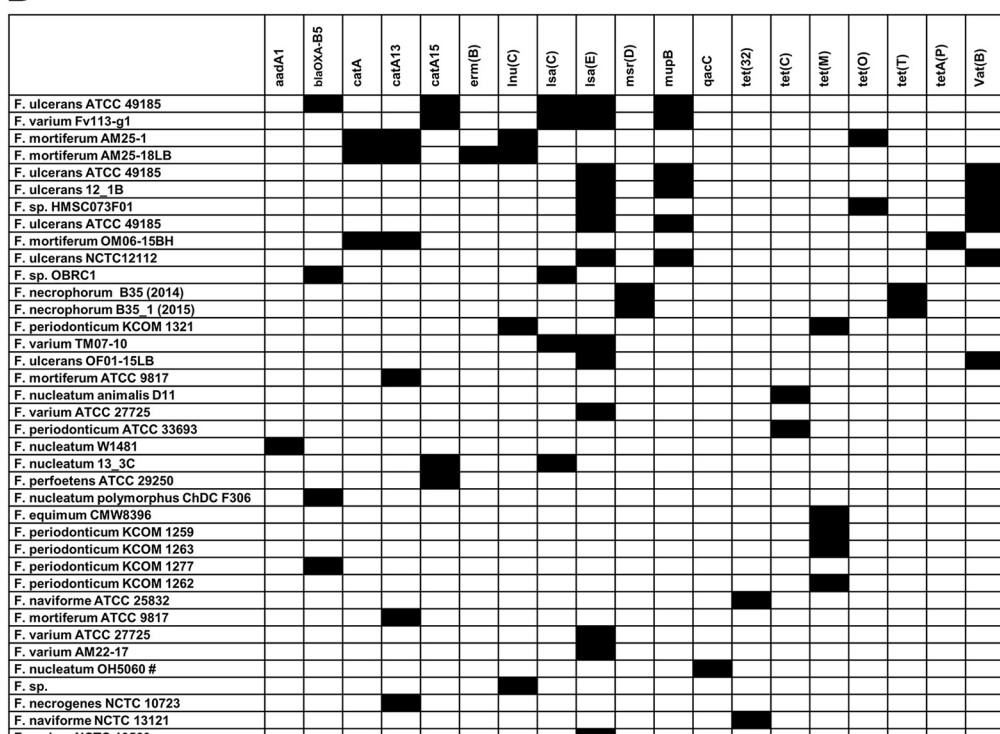

**B**

**FIG 3** (A) Virulence genes detected in *Fusobacterium necrophorum*. The heat map shows the percent sequence identity of 13 potential virulence genes compared to the 46 genes in the custom database. The names of *F. necrophorum* strains

genes were harbored by *F. ulcerans*, *F. varium*, *F. mortiferum*, *F. periodonticum*, *F. nucleatum*, *F. perfoetens*, *F. equinum*, *F. naviforme*, and *F. necrogenes*. However, *F. necrophorum* subsp. *funduliforme* B35 was the only *F. necrophorum* strain that carried ARGs.

We found the *lsaE* gene, which confers resistance to pleuromutilin, lincosamide, and streptogramin A, in 13 genomes of *F. ulcerans* and *F. varium* except for one genome (*Fusobacterium* sp. strain HMSC073F01). Also, we identified the presence of *vatB*, which confers streptogramin resistance, in *F. ulcerans*. Furthermore, *F. necrophorum* subspecies *funduliforme* B35 harbored both macrolide [*msr*(D)] and tetracycline [*tet*(T)] resistance genes.

**Recombination events in *F. necrophorum* core genes.** Recombination events can occur from one lineage to another within a single species and can lead to subspeciation. To investigate whether *F. necrophorum* subspeciation results from recombination events, we applied the ClonalFrameML model. Genome-wide recombination events in *F. necrophorum*, based on 46 genomes representing carriage and reference isolates assigned to *F. necrophorum* assembly genomes, are represented in Fig. 4A and B. Figure 4A is the graphical presentation of the recombination events, and Fig. 4B shows a phylogenetic tree constructed based on the recombination events. The dark blue line segments represent recombination sites. For a given branch, sites with no substitutions are shown in light blue, and any color ranging from white to red indicates substitution via mutation. Overall, we detected 1,615 recombination events.

Based on the ClonalFrameML analysis, we used the following metrics to study the relative effect of recombination and mutation on the *F. necrophorum* core genome (Data Set S8). This analysis yielded the following metrics: the relative rate of recombination with mutation ($R/\theta$) was 0.30455, the mean length of the recombined DNA ($\delta$) was 286, and the mean divergence of imported DNA ($\nu$) was 0.0371. Each recombination event generated approximately 10.61 substitutions (i.e., the product of $\nu$ and $\delta$). The ratio of changes introduced by recombination to changes introduced by mutation ($r/m$) was 3.226 (i.e., the product of $R/\theta$, $\delta$, and $\nu$) (19). These findings suggest that recombination caused three times more substitutions than mutation. Consistent with this, our pangenome analysis revealed minimal core gene similarity between *F. necrophorum* species, supporting our finding that there has been extensive genomic-level recombination. This is likely the case because core functions are less susceptible to recombination, which would confer a fitness disadvantage and result in the loss of bacteria from the population. This output suggests that recombination occurred in accessory genes associated with pathogenesis and other accessory functions. These accessory genes can undergo recombination without causing loss of function and may result in a divergent group within species. Extensive recombination events occurred in strain KG35 (Fig. 4A), and the recombination events commonly occurred among *F. necrophorum* subsp. *necrophorum* strains, giving rise to multiple evolutionary distances within the species.

Because recombination events are usually constructed to correct branch length and identify true phylogeny, we used recombination events analysis to determine whether the recombining genes and events support putative virulence character evolution in *Fusobacterium*. Our phylogenetic analysis (Fig. 4B) revealed that *F. necrophorum* subsp. *funduliforme* possibly underwent several recombination events to evolve into three distinct *F. necrophorum* subsp. *funduliforme* clades (clades I and III) and a group of uncharacterized *F. necrophorum* strains (clade II). Later, clade II evolved to form a distinct clade of *F. necrophorum* subsp. *necrophorum* (clade IIa), which includes test strains 8L1 and 212 and reference strains BL and FDAARGOS 565. When comparing the genomic size with respect to recombination, we found that the genomic size of the strains in clade II is

**FIG 3** Legend (Continued)

are shown on the right, and the virulence genes are below the heat map. Blue indicates ≤40%, white indicates 40 to 50%, and red indicates >50 to 100% sequence identity. Virulence gene acronyms are defined at the bottom. (B) ARGs detected in *Fusobacterium*. The gene matrix shows the presence and absence of genes detected in each *Fusobacterium* genome (indicated on the left) using ABRicate. Black indicates the presence and white indicates the absence of antimicrobial resistance genes shown at the tops of the columns. *aadA1*, aminoglycoside; *bla*OXA-85, beta-lactam; *catA*, *catA13*, *catA15*, chloramphenicol; *erm*(B), macrolide; *lnu*(C), lincosamide; *lsa*(C), ABC efflux/lincomycin, clindamycin, tiamulin; *lsa*(E), pleuromutilin, lincosamide, streptogramin A; *msr*(D), macrolide; *mupB*, mupirocin; *qacC*, quaternary ammonium compound; *tet*(32), *tet*(C), *tet*(M), *tet*(O), *tet*(T), *tetA*(P), tetracycline; *vat*(B), streptogramin.

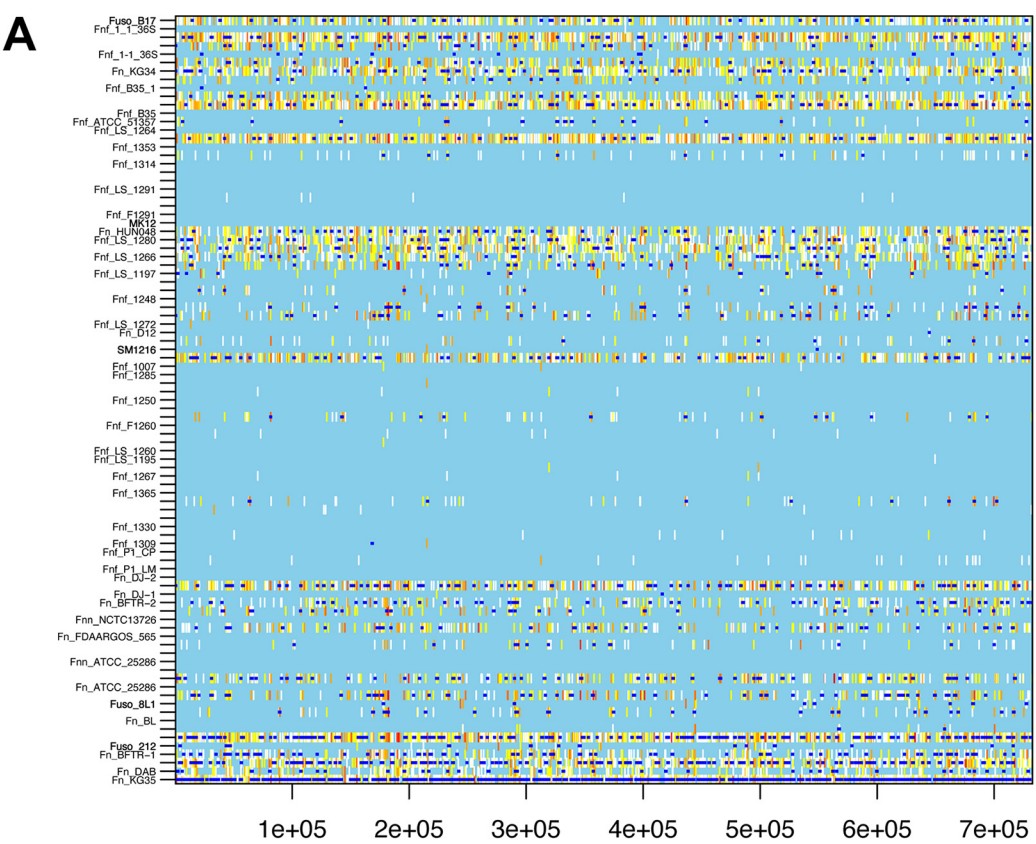

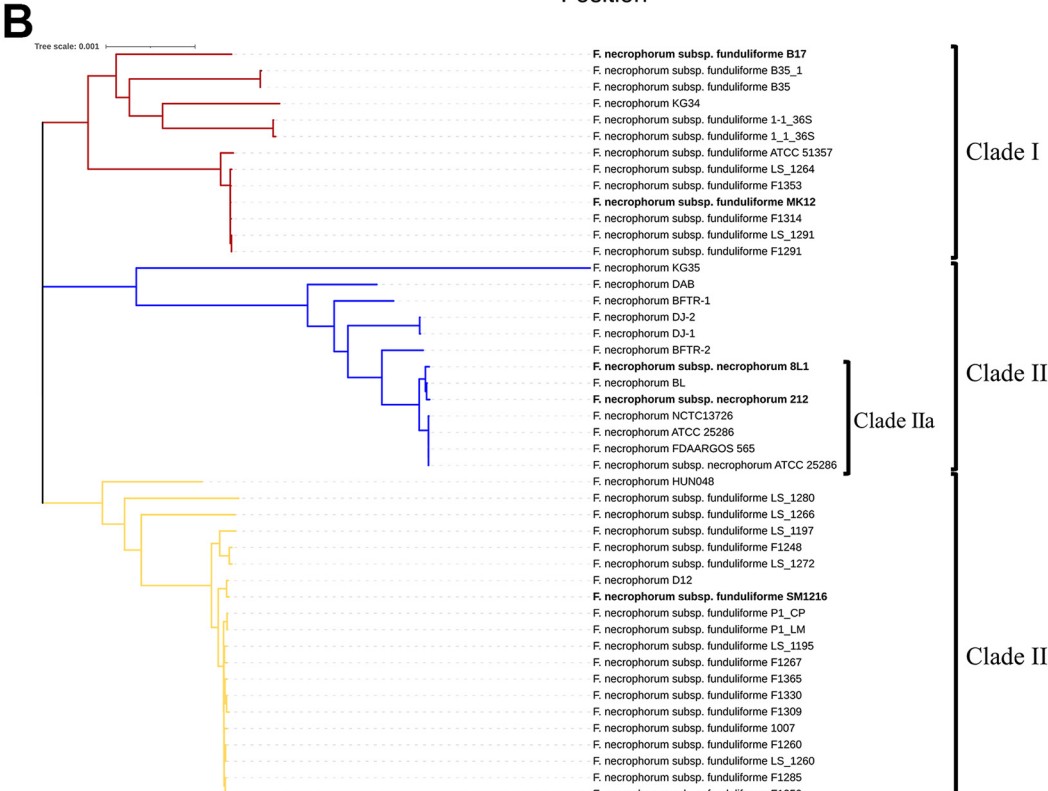

**FIG 4** (A) ClonalFrameML analysis of recombination in *F. necrophorum* based on 46 genomes mapped to *F. necrophorum* assembly genomes. In the recombination plot, white vertical bars indicate reconstructed substitutions and dark blue horizontal

greater than that of *F. necrophorum* subsp. *funduliforme* (Fig. 4B). These findings suggest that widespread recombination in clade II may give rise to subspecies different from *F. necrophorum* subsp. *funduliforme*. However, we did not identify a genomic size correlation with respect to strain KG35, which shows proximity in phylogeny-based recombination within clade II. Similarly, test strains MK12 and B17 appeared to have undergone fewer recombination events and lie closer to their ancestor strains, but each fell in a different branch, while SM1216 was in a different clade of *F. necrophorum* subsp. *funduliforme* descendants.

## DISCUSSION

In the present study, we provided genomic sequence analysis of the genus *Fusobacterium* and the species *F. necrophorum* to understand genomic variability and species relatedness and to characterize five clinical isolates. To date, very few studies have investigated the population structure of *Fusobacterium*. We included 167 *Fusobacterium* strains for which the whole genome has been sequenced (the 162 latest [July 2019] RefSeq strains of *Fusobacterium*, including *F. necrophorum* RefSeq strains, plus five test strains) for our analyses. The *Fusobacterium* phylogenic topology published by McGuire et al. is comparable to the phylogeny in our study (20), although *F. necrophorum* strain D12 was shown to be in closer relation to *F. gonidiaformans* ATCC 25563. However, the data comparison of this study was based on only 26 genomes. In our analysis, which consisted of 167 *Fusobacterium* genomes, we observed a similar output in our UBCG-based cladogram (Fig. 1A) and noted a close relationship of *F. necrophorum* with *F. gonidiaformans*. Furthermore, D12 is far inside with proximity to the test strain SM1216 in the cladogram of our study. Our analyses showed that *F. necrophorum* KG35 was the outermost genome, as this strain appeared to be an outlier in our subsequent analyses. This finding suggests that strain KG35 may be the most ancestral genome available for *F. necrophorum*, which could have descended from the species *F. gonidiaformans* during evolution. On the other hand, KG35 could belong to a different entity than *F. necrophorum*. This hypothesis could be corroborated by evaluating the ANI output for ANI values ranging below <95%, because 95% ANI-based demarcation is considered acceptable for intraspecies classification.

We used conserved genes or core genes to understand *Fusobacterium* species' phylogeny. Our analysis revealed a wide range of genomic variability, with only 29 core genes shared in the genus *Fusobacterium*, including those that encode ribosomal proteins and some essential catalyzing enzymes (Fig. 2A). The accessory genes contributed to >88% of the pangenome, thus indicating substantial heterogeneity among *Fusobacterium* species. All bacteria have developed ways to transfer or acquire genes through various mechanisms with high recombination rates (21). The variability we observed within this genus indicates that these bacterial species had been through a series of evolutionary events that gave rise to 28 species in the genus *Fusobacterium* (22), suggesting that these core genes may be essential for the survival of the bacteria.

The results of this study also explain the evolutionary changes that led to the genetic divergence of the species *F. necrophorum* into *F. necrophorum* subsp. *necrophorum* and *F. necrophorum* subsp. *funduliforme*. Critically, there were some *F. necrophorum* RefSeq strains used in this study that were not identified to the subspecies level. We were able to predict the subspecies of these unclassified RefSeq strains based on the constructed phylogenetic tree and how these strains clustered (Fig. 1B and Table 1). Therefore, our analyses predicted that strain BL and FDAARGOS 566 are *F. necrophorum* subsp. *necrophorum*.

**FIG 4** Legend (Continued)

bars indicate recombination events for each branch of the maximum-likelihood tree. The strain names are on the left, and the sizes and gene positions of the recombination events are shown across the alignment. (B) Phylogenetic tree constructed based on ClonalFrameML analysis of recombination in 46 *F. necrophorum* genomes, five tests and 41 reference strains. Three distinct clades are formed, with the red and yellow branches representing clade I and III, specific to *F. necrophorum* subsp. *funduliforme*, and the blue branch representing clade II, specific to *F. necrophorum* subsp. *necrophorum* (clade IIa), with *F. necrophorum* strains not identified to the subspecies level. Strain KG35 had the longest branch, which indicates significant recombination events. Test strains are in bold. The scale bar represents the length of the branch corresponding to the number of changes per site.

Additionally, our analyses indicated that the uncharacterized *F. necrophorum* strains DAB, BFTR-1, BFTR-2, DJ-1, and DJ-2 could belong to *F. necrophorum* subsp. *necrophorum*. Although the orthologous-group-based (pangenome) analysis confirmed genetic dissimilarity among subspecies, this analysis did not provide any definite correlation between GC content and source (23). Despite this, the genome was larger in *F. necrophorum* subsp. *necrophorum* than in *F. necrophorum* subsp. *funduliforme*.

To specify the genes unique to the subspecies, further analysis using pangenome output and the Scoary program identified 14 and 26 genes unique to the *F. necrophorum* subsp. *necrophorum* and *F. necrophorum* subsp. *funduliforme*, respectively. Among the unique genes specific to subspecies, no direct virulence gene was observed in any subspecies. However, different transferases, including those involved in vitamin biosynthesis that have a specific role in biochemical pathways and are integral to bacterial survival, were identified. Some vitamins are known to promote virulence, for example, vitamin $B_2$ and vitamin $B_6$ (24–26). Similarly, biosynthetic transferases for the vitamins cobalamin and thiamine identified in our analyses are also known to be involved in virulence and influence the ability of the pathogen to infect a host studied in some bacteria (27, 28). Some of the other transferases, such as 4-diphosphocytidyl-2-C-methyl-D erythritol kinase, which is involved in isoprene biosynthesis, could be involved in survival and competition in a mixed-culture environment and be specific to *F. necrophorum* subsp. *funduliforme* (29). Similarly, the iron transporter gene *hmu* is known to be involved in hemin utilization and is involved in iron acquisition in *Yersinia pestis* (30). The acquisition of iron is essential for bacterial growth and promotion, as well as for pathogenicity and virulence. This feature is specific to *F. necrophorum* subsp. *necrophorum*.

To evaluate genetic relatedness and characterize the five clinical isolates, we evaluated the ANI of orthologous fragment pairs shared between the two genomes (31). Our analysis revealed that the calculated ANI values between the *F. necrophorum* strains and the five test strains were ≥95% identical, a recommended threshold for demarcation for bacterial species (32). These results further confirmed that all test strains (8L1, 212, B17, MK12, and SM1216) belonged to *Fusobacterium necrophorum*. Kook et al. used the same approach to calculate ANI values in *F. nucleatum* subspecies and found that all four examined subspecies (*F. nucleatum* subsp. *nucleatum*, *F. nucleatum* subsp. *polymorphum*, *F. nucleatum* subsp. *vincentii*, and *F. nucleatum* subsp. *animalis*) had a pairwise ANI of >96% (33). The study by Kook et al. led to a result similar to that in Fig. 1A, where the phylogenetic tree based on the 16S rRNA gene was constructed and revealed that *F. hwasookii* clustered with *F. nucleatum* subsp. *polymorphum*. However, it has been argued that the above-mentioned subspecies of *F. nucleatum* should be considered separate species, as they all differ significantly at the DNA level (34).

With an increase in *Fusobacterium* pathogenesis, it is necessary to identify virulence genes that can be exploited to treat related diseases. The rapid rise of sequencing technologies and genomic analysis has provided an important platform for exploring causes of pathogenicity, as well as determining virulence genes and antimicrobial resistance patterns. However, the lack of reference genomes for more virulent subspecies, such as *F. necrophorum* subsp. *necrophorum*, for which few sequenced genomes are publicly available, has made phylogenetic analysis and identification of potential virulence factors a challenge. To overcome this challenge, we performed a custom-based virulence gene search using all 46 *F. necrophorum* strains, which enabled us to broaden the search probability. Based on this search, we identified 13 different virulence genes distributed among *F. necrophorum* strains. Some of these virulence genes were explicitly found in only one subspecies. For example, virulence genes encoding the type IV secretion system (T4SS), hemin receptor, and FadA adhesin were found only in the *F. necrophorum* subspecies *necrophorum*. On the other hand, YadA-like family proteins, hemagglutinin, OmpA, and Fic toxin, were only found in *F. necrophorum* subsp. *funduliforme*. Also, a set of genes, including *ompH* and those encoding virulence-associated protein (*vapD*), hemolysin III family protein, cell envelope integrity protein (*creD*), and leukotoxin, were found in both subspecies. However, there were

notable exceptions for *creD* and leukotoxin A genes, as *creD* was absent in *F. necrophorum* subsp. *funduliforme* F1365 and leukotoxin A was absent in *F. necrophorum* DJ1.

Our study found the *fadA* gene in *F. necrophorum* subsp. *necrophorum*. Because *fadA* is a virulence gene involved in host cell attachment and encodes a vital invasion factor for active invaders in *F. nucleatum* infections (35), our findings suggest that *F. necrophorum* subsp. *necrophorum* could be classified as an active invader (20, 36). The absence of the invasion factor (FadA) in *F. necrophorum* subsp. *funduliforme* could explain why *F. necrophorum* subsp. *funduliforme* often occur in association with mixed bacterial infection (37–39). However, more studies are needed to confirm whether *F. necrophorum* subsp. *necrophorum* is an active invader.

Previous reports have documented the role of T4SS in horizontal gene transfer of antimicrobial resistance and virulence genes (40, 41). Since *F. necrophorum* appeared to have evolved from *F. nucleatum*, the presence of T4SS in *F. necrophorum* subsp. *necrophorum* indicates the possibility of horizontal acquisition of *fad*A from *F. nucleatum* (42). Furthermore, the hemin receptor (hemin_rec), which was found in *F. necrophorum* subsp. *necrophorum* (43), plays an essential role in the colonization and proliferation of many pathogenic bacteria. Therefore, the hemin receptor may be a potential target for the development of vaccines and therapies for *F. necrophorum* infections.

Fusobacterial infections are commonly treated with antibiotics such as penicillin, amoxicillin, clindamycin, and imipenem. Although antimicrobial resistance is not commonly found in this genus, in contrast to other pathogens, the emergence of resistance is inevitable. Our genome sequence analysis showed few ARGs distributed among all *Fusobacterium* species other than *F. necrophorum*. Only two ARGs were found in *F. necrophorum* (1/46 strains tested), indicating the chances of antimicrobial resistance emergence in *Fusobacterium* species. Therefore, it is necessary to develop novel and effective therapeutic and prevention strategies, such as vaccination, before the development of resistance in this bacterial species. It is of the utmost importance to search for potential virulence targets that could be used to develop therapeutic and prevention strategies against fusobacterial infection.

Pathogenic and pandemic fusobacterial strains are often the result of recombination events. We focused this study on *F. necrophorum* subsp. *necrophorum* and *F. necrophorum* subsp. *funduliforme*, as their biological activities differ. We investigated recombination events that might have influenced evolutionary diversification of the *F. necrophorum* subspecies. The results revealed that *F. necrophorum* subsp. *funduliforme* may have undergone a genomic expansion during evolution, leading to a clade of a virulent form of *F. necrophorum* subsp. *funduliforme* and another unclassified clade of *F. necrophorum*. Later, this unclassified *F. necrophorum* subsp. *necrophorum* clade could have evolved to more virulent forms of *F. necrophorum* subsp. *necrophorum*. Similarly, some of the virulence genes found only in *F. necrophorum* subsp. *necrophorum* reflect evidence of recombination during evolution in the *F. necrophorum* subsp. *necrophorum* genome. Our data demonstrate that the difference observed is evidence for the effect of recombination events on the evolutionary relationship with other *Fusobacterium necrophorum* strains (Fig. 4A) (44). Taken together, the results of this study suggest that recombination events played a significant defining role in *F. necrophorum*, resulting in *F. necrophorum* subsp. *necrophorum* being more virulent than *F. necrophorum* subsp. *funduliforme*.

## MATERIALS AND METHODS

**Strains.** Five strains of *F. necrophorum*, including four bovine isolates (8L1, B17, 212, and SM1216) and one human isolate (MK12), were used in this study (Table 1). These strains were isolated and identified from previously described clinical cases (37). For all strains, DNA extraction was performed as previously mentioned (45). Briefly, the strains were grown in prereduced anaerobically sterilized brain heart infusion (PRAS-BHI) broth. After overnight growth, cells were pelleted by centrifugation (6,500 × *g*) and resuspended in TES buffer (50 mM Tris-HCl [pH 7.5], 1 mM EDTA, and 25% sucrose). The cells were then pretreated with lysozyme (room temperature, 30 min) and lysed using Sarkosyl NL (Sigma-Aldrich, St. Louis, MO, USA) with proteinase K (Sigma-Aldrich) (60°C, for 1 h). The crude lysate was resuspended in Tris-buffer-saturated phenol-chloroform. DNA was extracted by precipitation in 2.5 mL ice-cold ethanol, and the pellet was resuspended in TE buffer (10 mM Tris-HCl and 1 mM EDTA; pH 8). Subsequently, the DNA was purified with a cesium chloride density gradient followed by dialysis against double-distilled

water. The purity and concentration were checked spectrophotometrically. Purified DNA was used for paired-end sequencing at HiSeq using the Illumina platform.

Furthermore, 162 *Fusobacterium* sequence files and 41 *F. necrophorum* RefSeq assemblies were downloaded from the NCBI database (Data Sets S1 and S2). The sequences downloaded for this study were selected to generate a diverse data set for comparative genome analysis. Furthermore, genomes of strains belonging to two different genera, *Leptotrichia buccalis* DSM 1135 and *Cetobacterium somerae* ATCC BAA-474, were included in the data as an outgroup.

**Bioinformatic analysis. (i) Data preprocessing.** All raw read preprocessing was performed on the Galaxy platform (https://usegalaxy.org/). For quality control, raw sequences were preprocessed using FastQC software v0.11.7. Based on the FastQC report, quality trimming was performed using Trimmomatic to filter and trim low-quality reads (46). The processed paired-end reads were used for subsequent analysis.

**(ii) Genome assembly, quality check, and annotation.** The quality trimmed reads were assembled on the Galaxy platform (https://usegalaxy.org/) using SPAdes with default parameters (47), and contigs shorter than 200 bp were discarded. The quality of the final assemblies was checked using QUAST, a quality assessment tool (48). Using the Rapid Prokaryotic Genome Annotation (PROKKA) annotation, curated assemblies were annotated (49). Subsequently, the annotation files (.gff) produced by PROKKA analysis were used to create the pangenome.

**Phylogenetic analysis by core gene alignment and average nucleotide identity.** Phylogenetic trees were constructed based on the core gene alignment (UBCG) and the average nucleotide identity (ANI) values.

**(i) Up-to-date bacterial core gene.** To predict genomic relationships at the genus and species levels, a UBCG approach with default parameters was used (50). These trees were constructed using the concatenated alignment of the core genes of bacteria. A manually annotated GenBank file with 162 NCBI reference *Fusobacterium* sequences and the latest NCBI RefSeq assembly of 41 *F. necrophorum* species were downloaded (https://www.ncbi.nlm.nih.gov/assembly; last accessed July 2019). The sets of 167 *Fusobacterium* genus genomes (162 NCBI RefSeq sequences plus five test strains) were used for the analysis of evolutionary relationships at the genus level between different species. A similar phylogenetic analysis was performed for 46 (41 NCBI RefSeq sequences plus five test strains) *F. necrophorum* genomes. For each UBCG, the core gene sequences were aligned with multiple-sequence alignment. The resulting alignments were concatenated, and phylogeny was inferred using the maximum-likelihood (ML) algorithm in the RAxML v 7.8.6 tool using the GTR+I+G4 model of nucleotide substitution (51). All trees were visualized using iTOL, a web tool for phylogenetic tree display (https://itol.embl.de/) (52) (Data Set S3).

**(ii) Average nucleotide identity analysis.** The DNA-DNA relatedness and orthologous genes shared between species of *F. necrophorum* were analyzed using the FastANI algorithm by constructing an ANI heat map (53). Genetic relatedness for pairwise comparisons of 46 genome sequences, including five test strains, was measured by the ANI of all conserved orthologous genes calculated using the FastANI method (version 1.3). ANI values were calculated as previously described (53). Items in the FastANI matches that showed ≥95% nucleotide sequence identity and ≥70% sequence coverage were selected to calculate ANI. The fragment length was set at 1,000 bases and a k-mer size of 16, with the minimum fraction set to 0.5. The output was used to generate a heat map and identify relatedness.

**Pangenome analysis.** Pangenome analyses are composed of core genome detection (genes common to all strains) and detection of accessory or ancillary genes (genes common to only a subset of the strains and strain-specific genes). Comparative genomic analysis was performed for the five curated genome assemblies using the published NCBI RefSeq database for *Fusobacterium* and *F. necrophorum* at both the genus and species levels. Genome sequences annotated by the PROKKA (49) pipeline using the Roary program (54) were used to create the pangenome. The core gene alignment file obtained from Roary was taken for model testing using RAxML. Based on RAxML, the GTR+I+G4 model was the best-scoring model to generate the ML estimate. Later, this model was used to generate the phylogenetic trees; ML and iTOL were used for phylogenetic tree construction.

A pairwise comparison was performed to identify subspecies specific gene differences between *F. necrophorum* subspecies. The comparison was performed in the Scoary program to identify the uniqueness of the gene via evaluation of the highest-scoring nonintersecting gene pairs, sorted by *P* value (55). The analysis was done using the gene presence-absence output of Roary and UBCG subspecies classification as phenotypic traits.

**Identification of virulence genes.** We methodically searched the literature to identify potential virulence genes in *Fusobacterium* and related species for virulence genes. The literature search results were used to create a list of potential virulence genes; then, a local database was created through a manual search with the FASTA sequences of these genes (Data Set S5). These genes were checked against the strains used for comparative analysis at the genus and species levels.

**Identification of antimicrobial resistance genes.** ARGs were investigated in the latest NCBI RefSeq sequences of the genus *Fusobacterium* and the contigs of the five test strains. The ABRicate v0.8.10 search engine (https://github.com/tseemann/ABRicate) was used to screen ARGs with the following databases: ResFinder (56), NCBI AMR FinderPlus (57), and CARD (v2.0.3) (58). Positive-hit cutoffs of 50% sequence identity and 30% sequence coverage were used for analysis.

**Recombination analysis.** Recombination events and evidence of positive selection were analyzed using ClonalFrameML, an interactive application using default parameters (59). Briefly, multiple-sequence alignments of RefSeq *F. necrophorum* genomes with test strains were inferred using RAxML and analyzed using ClonalFrameML. An ML tree was constructed based on the recombination events, and the clonal genealogy was reiterated until the best tree was obtained. ClonalFrameML uses Baum-Welch expectation maximization

and Viterbi algorithms to obtain an ML estimate of the recombination parameters, branch lengths of clonal genealogy, and ML importation status.

**Data availability.** Sequencing raw data are available from the latest version of RefSeq: https://www.ncbi.nlm.nih.gov/assembly/?term=Fusobacterium and https://www.ncbi.nlm.nih.gov/assembly/?term=Fusobacterium±necrophorum. Assembly accession numbers (or GenBank Assembly IDs and RefSeq Assembly IDs) for the assemblies analyzed in this manuscript are provided in the supplementary material see Data Set S1.

## SUPPLEMENTAL MATERIAL

Supplemental material is available online only.

**SUPPLEMENTAL FILE 1**, XLSX file, 0.03 MB.
**SUPPLEMENTAL FILE 2**, XLSX file, 0.01 MB.
**SUPPLEMENTAL FILE 3**, XLSX file, 0.01 MB.
**SUPPLEMENTAL FILE 4**, XLSX file, 1.7 MB.
**SUPPLEMENTAL FILE 5**, XLSX file, 0.01 MB.
**SUPPLEMENTAL FILE 6**, XLSX file, 0.01 MB.
**SUPPLEMENTAL FILE 7**, XLSX file, 0.01 MB.
**SUPPLEMENTAL FILE 8**, XLSX file, 0.1 MB.
**SUPPLEMENTAL FILE 9**, PDF file, 0.2 MB.

## ACKNOWLEDGMENTS

We are thankful to the team at South Dakota State University, where all bioinformatic coding was performed for this study. The clinical test strains were isolated at Kansas State University as part of a research study.

We declare no conflict of interest.

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
