## [Reviewer comments · Microbiology Spectrum]

Microbiology Spectrum

Comparative Genomic Analysis of *Fusobacterium necrophorum* Provides Insights into Conserved Virulence Genes

Prabha Bista, Deepti Pillai, Chayan Roy, Joy Scaria, and Sanjeev Narayanan

Corresponding Author(s): Prabha Bista, Purdue University

Review Timeline:

Submission Date:	January 25, 2022
Editorial Decision:	March 11, 2022
Revision Received:	June 13, 2022
Editorial Decision:	July 8, 2022
Revision Received:	September 13, 2022
Accepted:	September 16, 2022

Editor: Philip Rather

Reviewer(s): The reviewers have opted to remain anonymous.

Transaction Report:

DOI: <https://doi.org/10.1128/spectrum.00297-22>

March 11, 2022

Dr. Sanjeev K Narayanan
Purdue University
Department of Comparative Pathobiology
102 VPTH, 725 Harrison Street, West Lafayette, Indiana 47907, USA
West Lafayette, IN 47907

Re: Spectrum00297-22 (Comparative Genomic Analysis of *Fusobacterium necrophorum* Provides Insights into Conserved Virulence Factors)

Dear Dr. Sanjeev K Narayanan:

Thank you for submitting your manuscript to Microbiology Spectrum. The manuscript was reviewed by two experts in the field. One reviewer did not find merit to the study and the other reviewer raised a number of concerns that need to be addressed. If you feel that you can address these concerns, I will consider a revised manuscript. This revision may be sent out for a second review.

Link Not Available

Sincerely,

Philip Rather

Journals Department
Reviewer comments:

Reviewer #1 (Comments for the Author):

The article of Bista et al investigate the pangenome of *Fusobacterium necrophorum*, a cattle pathogen. This is a completely in silico project using pre-existing mostly open source software tools. The new data here is the genome sequence of 5 strains and they use 162 genome sequences from public databases. There is no description of the quality of the new data (or the existing

data). The reason for choosing to sequence these 5 strains is not explained and, in fact, the analysis and conclusions in the article could have been performed without their inclusion. There is no obvious hypothesis to the work. The manuscript proceeds through a series of descriptive bioinformatic analyses: pangenome, ANI, antibiotic resistance genes etc. Perhaps the most interesting finding is the absence of the possible absence of the FadA virulence factor in the funduliforme subspecies but the number of sequenced strains from the *Fusobacterium necrophorum* is quite low and this might be a feature of small sample size. The major result reported by the authors, the *Fusobacterium* genus has "0.05 matched core genes" is just an artifact of the thresholds used in the ROARY program, which was designed for within-species rather than within-genus analysis. It is concerning that there is an apparent misunderstanding of the pangenome concept here, which is central to the paper. Overall, the paper was difficult to follow, with many confusing passages and interpretations of data that were questionable. The figures were of low quality and the legends inadequate. There is little new data and the results are of quite low significance to the field.

As a afternote - the acknowledgements section lists Dr Charyan Roy who "performed the bioinformatics coding for this study and provided us with the result outputs". They should be listed as co-author on this paper for that fundamental contribution.

Reviewer #2 (Comments for the Author):

General Comments

The manuscript reports on the genomic analyses of *Fusobacterium* species, mainly *F. necrophorum*. Because *Fusobacterium*, except for the human oral pathogen, *F. nucleatum*, is somewhat under investigated Gram negative anaerobic pathogen, there is merit to the study reported in this manuscript. The main focus of the study appears to be comparative analysis of the two subspecies of *F. necrophorum*, however, the inclusion of other species of *Fusobacterium* has somewhat diluted the importance of the study. Because the subsp. *funduliforme* is the one involved in human infections, authors should make a comparative analysis between human and animal strains of the subspecies.

Authors have used the terms virulence genes and virulence factors almost interchangeably, while the study is only on the virulence genes, and not the virulence factors.

Title

The study is on the analysis of virulence genes, not the virulence factors. The title should be changed accordingly

Line 11: Is this sentence on the genus *Fusobacterium* or the species *F. necrophorum*?

Line 15: You should identify the five isolates as four bovine and one human

Line 26: Virulence factors or virulence genes? The analysis is on the virulence genes, not the virulence factors here and in the rest of the manuscript.

Line 47: You should include tonsillitis, which is much more common than Lemierre's syndrome

Line 59: should be antimicrobial resistance

Line 63: Should be virulence, not pathogenesis

Lines 69-70: the MK1 is a human strain!

Lines 83-85: You may want to include a table that lists all the species and the number of strains in each used in the study.

Line 158: *Fusobacterium* genus

Line 162-163: Are these represented in Figure 1A?

Lines 171-175: This should be in the discussion section

Line 181: How was this subspeciation done with the WGS data?

Line 193: You should give mean and or range in ANI values between the two subspecies of *F. necrophorum* and also compare between bovine and human strains

Line 205: The description of pan- and core- genome analyses should include comparison between the two subspecies of *F. necrophorum*.

Line 207: It may be useful to some description of the core 29 genes.

Line 227: Description of the virulence genes should come before ARGs

Line 228: Why call it acquired?

Lines 230-231: in 38? Does it mean the rest of the 100 plus strains did not carry ARG?

Line 231; ARG or AMR. Antibiotic or Antimicrobial? Why not pick one and use it consistently

Line 239: 12 genomes of varium and ulcerans?

Line 247-248: These virulence genes, not factors

Lines 326-331; No need to repeat the results

Lines 348-350: can you make the same argument for the two subspecies of necrophorum?

Figure 1A: The legend should include the red color

Table 1: This may be deleted or include this in Table 2

Table 2: You may want to add a column to identify the source, host, clinical or otherwise

Staff Comments:

Preparing Revision Guidelines

Please return the manuscript within 60 days; if you cannot complete the modification within this time period, please contact me. If you do not wish to modify the manuscript and prefer to submit it to another journal, please notify me of your decision immediately so that the manuscript may be formally withdrawn from consideration by Microbiology Spectrum.

Reviewer #1 (Comments for the Author):

The article of Bista et al investigate the pangenome of *Fusobacterium necrophorum*, a cattle pathogen. This is a completely in silico project using pre-existing mostly open source software tools. The new data here is the genome sequence of 5 strains and they use 162 genome sequences from public databases.

There is no description of the quality of the new data (or the existing data).

- We understand the reviewer's concern regarding the quality of data. The quality of genome sequence of these strains was tested through FastQC software and proceeded for further analyses only after meeting high quality standard (described in the manuscript). We only presented the five sequences that passed the quality scoring cutoff of FastQC and not others. We can provide the quality control report if needed.

The reason for choosing to sequence these 5 strains is not explained. In fact, the analysis and conclusions in the article could have been performed without their inclusion. There is no obvious hypothesis to the work.

- Complied with the comments, the reasonings are explained in the text in line 61 and onwards of the revised manuscript. Thank you.

The manuscript proceeds through a series of descriptive bioinformatic analyses: pangenome, ANI, antibiotic resistance genes etc. **Perhaps the most interesting finding is the absence the possible absence of the FadA virulence factor in the funduliforme subspecies but the number of sequenced strains from the *Fusobacterium necrophorum* is quite low and this might be feature of small sample size.**

- Based on the comment, the presence of FadA in subspecies *necrophorum* and absence in subspecies *funduliforme* has been discussed in line 392-396.

The major result reported by the authors, the *Fusobacterium* genus has "0.05 matched core genes" is just an artifact of the thresholds used in the ROARY program, which was designed for within-species rather than within-genus analysis. It is concerning that there is an apparent misunderstanding of the pangenome concept here, which is central to the paper. **Overall, the paper was difficult to follow, with many confusing passage and interpretations of data that were questionable. The figures were of low quality and the legends inadequate.**

- We have tried to address this comment as much as possible to improve the flow of the manuscript, eliminate ambiguity and to focus on the major conclusions of the study. The figure quality has been improved and figure legends have been updated.

There is little new data and the results are of quite low significance to the field.

- Complied with the comment, it is true that there is little new data and we appreciate the reviewer's concern. However, very limited genome sequences are available for the *F. necrophorum*. Therefore, it is our attempt to add more genome sequences of *F. necrophorum* to the public database. Since there is limited data available, we have very limited information on this opportunistic pathogen, *F. necrophorum*, which has significant impact on feedlot industry. Hence, we attempt to gather information on the bacteria through genomics approach, a study that was not provided earlier regarding this particular species.

As a afternote - the acknowledgements section lists Dr Charyan Roy who "performed the bioinformatics coding for this study and provided us with the result outputs". They should be listed a co-author on this paper for that fundamental contribution.

- Dr. Chayan Roy has been added as a co-author to the manuscript, consulted for his thoughts on the reviewers' comments and helped drafting the revised manuscript.

Reviewer 2: General Comments

The manuscript reports on the genomic analyses of *Fusobacterium* species, mainly *F. necrophorum*. Because *Fusobacterium*, except for the human oral pathogen, *F. nucleatum*, is somewhat under investigated Gram negative anaerobic pathogen, there is merit to the study reported in this manuscript. The main focus of the study appears to be comparative analysis of the two subspecies of *F. necrophorum*, however, **the inclusion of other species of *Fusobacterium* has somewhat diluted the importance of the study. Because the subsp. *funduliforme* is the one involved in human infections, authors should make a comparative analysis between human and animal strains of the subspecies.**

- We understand the reviewer's concern, however, subsp. *funduliforme* was also isolated and observed in animal infections. This analysis did not follow the definite pattern based on host/source. We have tried to construct the population structure cladogram and analyze them based on source (Figure 2B) which did not give a definite correlation as mentioned in line 360 of the discussion section of the revised manuscript.

Authors have used the terms virulence genes and virulence factors almost interchangeably, while the study is only on the virulence genes, and not the virulence factors.

- Complied with the comment, this has been corrected in the revised manuscript, we only used the term "virulence genes", thank you.

Title

The study is on the analysis of virulence genes, not the virulence factors. The title should be changed accordingly

- The title has been corrected and the term "virulence genes" has been written in place of "virulence factors", thanks.

Line 11: Is this sentence on the genus *Fusobacterium* or the species *F. necrophorum*?

- Based on the comment, the abstract was edited to explain the discussion of species *F. necrophorum*.

Line 15: You should identify the five isolates as four bovine and one human

- Complied with this comment, the five isolates were described as four bovine and one human in the text (line 17 and 82)

Line 26: Virulence factors or virulence genes? The analysis is on the virulence genes, not the virulence factors here and in the rest of the manuscript.

- The phrase "virulence genes" has been corrected in the entire manuscript.

Line 47: You should include tonsillitis, which is much more common than Lamierre's syndrome

- Complied with the comment, the localized infection tonsillitis has been included in the manuscript (line 51)

Line 59: should be antimicrobial resistance

- The word "antimicrobial" has been added in the manuscript (Line 71), thanks.

Line 63: Should be virulence, not pathogenesis

- The term "virulence" instead of "pathogenesis" has been corrected (line 77), thanks

Lines 69-70: the MK1 is a human strain!

- Complied with the comment, MK12 has been described as a human strain in the manuscript, line 83, thanks

Lines 83-85: You may want to include a table that lists all the species and the number of strains in each used in the study.

- The lists of all the species and strains used in the analysis with their metadata has been submitted as supplementary files (S1 and S2)

Line 158: Fusobacterium genus

- The word "genus" has been corrected in revised manuscript (Line 173)

Line 162-163: Are these represented in Figure 1A?

- Outgroups are represented in red font and complied to the comment it is explained in the legend section

Lines 171-175: This should be in the discussion section

- In accordance to this comment, this paragraph as has been removed and explained in the discussion section in the revised manuscript (Line 329- 335)

Line 181: How was this subspeciation done with the WGS data?

- Complied with the comment, the subspeciation based on UBCG analysis has been mentioned and explained in the manuscript (Line 190)

Line 193: You should give mean and or range in ANI values between the two subspecies of F. necrophorum

- Complied with the comment, the range of ANI values between two subspecies has been mentioned in the manuscript (Line 206).

and also compare between bovine and human strains

We understand the reviewer's concern on comparison between bovine and human strains but since the source of *F. necrophorum* varies from bovine, human, environmental, deer to others. And with the limited data available, no such comparison was done for ANI analysis. Instead we used ANI to confirm the belongingness of test strains to *F. necrophorum* as described in line 212 of the revised manuscript.

Line 205: The description of pan- and core- genome analyses should include comparison between the two subspecies of *F. necrophorum*.

- We understand the reviewer's concern but the confirmed complete genome available so far for subspecies *necrophorum* is only one i.e *F. necrophorum* subspecies *necrophorum* ATCC 25286. This study proposed multiple subspecies *necrophorum* based on the study however still with limited data, the comparison study was not feasible and was not performed. However, pangenome analysis among species *F. necrophorum* is done and output is discussed (Line 229-241). Thank you.

Line 207: It may be useful to some description of the core 29 genes.

-The lists of these core genes are provided in supplementary file (S7). Complied to the comment, these genes mostly included ribosomal protein and other enzyme coding genes and mentioned in the manuscript (Line 226)

Line 227: Description of the virulence genes should come before ARGs

-Complied with this comment, this has been corrected in the entire manuscript including methodology and result section, thanks.

Line 228: Why call it acquired?

- The word acquired has been removed in the manuscript and reported as "Antimicrobial Resistance Genes (ARGs)" (Line 267), thanks.

Lines 230-231: in 38? Does it mean the rest of the 100 plus strains did not carry ARG?

-Yes, only 38 genomes out of 167 strains showed to be carried the ARGs available in ABRicate databases.

Line 231; ARG or AMR. Antibiotic or Antimicrobial? Why not pick one and use it consistently

- Complied with the comment, ARGs (Antimicrobial) has been used in the entire manuscript.

Line 239: 12 genomes of *varium* and *ulcerans*?

-Yes, 13 genomes of *F. varium* and *F. ulcerans* has *IsaE* gene. This has been mentioned in line 278 and output has been submitted as a supplementary figure.

Line 247-248: These virulence genes, not factors

-The word "factors" has been replaced with "genes" in the entire manuscript.

Lines 326-331; No need to repeat the results

- Complied with the comment, the repetition of results has been removed.

Lines 348-350: can you make the same argument for the two subspecies of *necrophorum*?

-The analysis shows the two subspecies belong to the species *necrophorum*. Though the analyses show they diverge in the genomic content, however, as mentioned previously the lack of enough data makes it not possible to interpret such argument.

Figure 1A: The legend should include the red color

-In accordance to the comment, the indication of outgroup in red color has been described in the figure legend.

Table 1: This may be deleted or include this in Table 2

Table 2: You may want to add a column to identify the source, host, clinical or otherwise

- Table 1 has been removed and a new table with source has been generated.

July 8, 2022

Mx. Prabha Kiran Bista
Purdue University
Department of Comparative Pathobiology
625 Harrison Street
West Lafayette, IN 47907

Re: Spectrum00297-22R1 (Comparative Genomic Analysis of *Fusobacterium necrophorum* Provides Insights into Conserved Virulence Genes)

Dear Mx. Prabha Kiran Bista:

Your revised manuscript has been evaluated by one of the previous reviewers. They still had several concerns that will need to be carefully addressed. In addition, please have your manuscript carefully proofread. I will not recommend acceptance until these concerns are addressed.

Reviewer's comments: As I read the revised manuscript, authors have complied with many of comments, except the major one, which was to focus on the species and subspecies of *F. necrophorum*. All other comments of mine were minor, more editorial. Also, I do not think authors have addressed the comment on the hypothesis. I still believe there is merit in the study reported in this manuscript. But it is poorly written and would require extensive editing to make it acceptable.

Link Not Available

Sincerely,

Philip Rather

Journals Department
Reviewer comments:

Staff Comments:

Preparing Revision Guidelines

Please return the manuscript within 60 days; if you cannot complete the modification within this time period, please contact me. If you do not wish to modify the manuscript and prefer to submit it to another journal, please notify me of your decision immediately so that the manuscript may be formally withdrawn from consideration by Microbiology Spectrum.

As I read the revised manuscript, authors have complied with many of comments, except the major one, which was to focus on the species and subspecies of *F. necrophorum*. All other comments of mine were minor, more editorial. Also, I do not think authors have addressed the comment on the hypothesis. I still believe there is merit in the study reported in this manuscript. But it is poorly written and would require extensive editing to make it acceptable.

- We had previously attempted to observe subspecies level difference in pangenome output through isolation source/host, GC%, and genome size. But these phenotypes did not show any definite classification or distinction among subspecies. Therefore, in accordance with the reviewers' comments on focusing on subspecies of *F. necrophorum*, we attempted to identify specific genes of the subspecies of genes mentioned in the manuscript (Line 25-27, line 125-163, line 255-265 and Line 389-403) .
- The hypothesis on the manuscript has been revised and rewritten in the Introduction section (lines 67-78).
- In terms of manuscript writing, major changes have been highlighted, and we have tried to translate some of the writing into a more understandable format.

September 16, 2022

Mx. Prabha Kiran Bista
Purdue University
Department of Comparative Pathobiology
625 Harrison Street
West Lafayette, IN 47907

Re: Spectrum00297-22R2 (Comparative Genomic Analysis of *Fusobacterium necrophorum* Provides Insights into Conserved Virulence Genes)

Dear Mx. Prabha Kiran Bista:

Your manuscript has been accepted, and I am forwarding it to the ASM Journals Department for publication. You will be notified when your proofs are ready to be viewed.

Sincerely,

Philip Rather
Editor, Microbiology Spectrum

Journals Department
Supplemental Material 3: Accept
Supplemental Material 4: Accept
Supplemental Material 7: Accept
Supplemental Material 5: Accept
Supplemental Material 2: Accept
Supplemental Material 1: Accept
Supplemental Material 6: Accept
Supplementary figure 1: Accept
Supplemental Material 8: Accept